# To guide or not to guide: improving diffusion sampling with progressive guidance

## Abstract

Diffusion guidance pertains to the use of conditional diffusion under specific constraints, such as class labels or textual inputs. There are primarily two methodologies: *Classifier Guidance* leverages a pre-trained classifier to guide generation towards the condition, while *Classifier-Free Guidance* achieves implicit guidance without any classifier but by entering the condition during training. Both approaches employ a weighting parameter $\omega$ that determines the trade-off between sample fidelity (unconditional diffusion) and conditional adherence. In this paper, we posit that such conflict between image quality and condition arises, in part, due to misclassification and conflicted gradients from the explicit or implicit classifier, especially when the noise is high i.e., at the first stages of generation. To address this, we introduce a progressive weighting scheme, called *Progressive-Guidance*, where we make the weight of the guidance term dependent on the timestep. We propose two-time dependent weighting schemes: a simple heuristic, and a more precise gradient-norm-based method. *Progressive-Guidance* can be implemented without retraining the model and with only a few additional lines of code. We report enhanced performance in benchmark metrics on three tasks: class-conditional image generation, text-to-image generation, and text-to-motion generation.

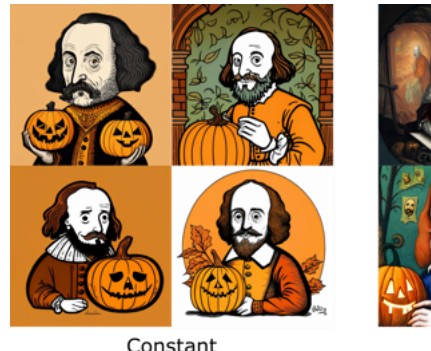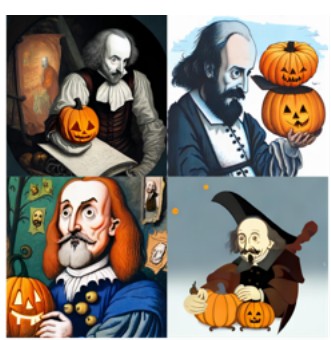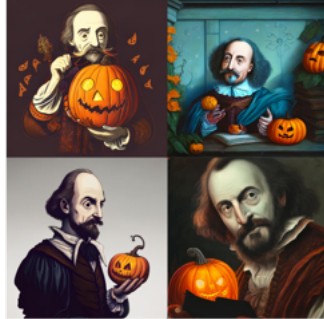

Figure 1: Our proposed progressive guidance schedules improve the sampling of diffusion models. (left) When sampling with a constant scheduler, the resulting images are similar (low diversity). On the contrary, when we adopt either (middle) a heuristic cosine scheduler guidance or (right) a grad-norm based once, we observe diversity in style, colors and backgrounds. Prompt: "A colorful cartoon of William Shakespeare philosophically looking at a jack-o-lantern in his hand".

## 1 Introduction

Diffusion models are a large family of generative models, demonstrating prominent generative capabilities in terms of output quality and conditioning flexibility. They are trained to transform a simple distribution, typically Gaussian, to a complex data distribution through the progressive removal of noise, leading to impressive generations across various domains e.g. images Ho & Salimans (2022), videos Luo et al. (2023), acoustic signals Kang et al. (2022), or even 3D avatars Chen et al. (2023).

Conditional generation (e.g. text-conditioned image generation) with diffusion models has been explored in numerous works (Saharia et al., 2022; Ruiz et al., 2023; Balaji et al., 2022), and is achieved in its simplest form by adding an extra input to the model, typically with residual connections as in Nichol & Dhariwal (2021). To enforce condition reliance of the model, classifier guidance (Dhariwal & Nichol, 2021) was proposed where the gradients of a separately trained, noise-dependent image classifier are linearly combined with those of a diffusion model. Alternatively, Ho & Salimans (2021) introduce a discrepancy mechanism between the conditional and the unconditional output of a previously trained diffusion model to better highlight the reliance on the condition, without the need for an external classifier. In both cases, a weighting parameter $\omega$ is introduced to control the weight of the generative term versus the guidance term and is directly applied to all timesteps via scalar multiplication. Varying $\omega$ is a trade-off between fidelity and condition reliance, as an increase in condition reliance often results in a decline in fidelity and also diversity.

In practice, combining such weighting mechanisms with the diffusion objective leads to some flaws during generation. For instance, (Dinh et al., 2023) highlight the presence of conflicting gradients between generation and guidance terms, especially during the early inference stages, and introduce a gradient correction approach to counteract this. From the perspective of diffusion for classification, Li et al. (2023) reveal a timestep discrepancy with respect to performance, i.e. using initial timesteps for classification results in a lower accuracy compared to intermediary timesteps. Choi et al. (2022) propose an empirically parameterized curve to allocate a higher training loss weight at certain timesteps to achieve more appealing results. Chang et al. (2023) notice that increasing the guidance scale linearly and replacing the unconditional term with a Negative Prompt enhances diversity. Yet, their changes were empirical without deep exploration or thorough evaluation. With a similar observation, Gao et al. (2023) create a parameterized power-cosine-like curve and optimize a dedicated parameter for their own dataset and method.

Despite the empirical solutions, no work provides a thorough or systematic investigation of the diffusion guidance behaviour at different timesteps. To bridge this gap, in this paper we delve into diffusion guidance behaviour and systematically examine how the condition influences the generation process. Our findings suggest that the influence of the condition is not uniformly distributed across all timesteps. Rather, certain timesteps are more critical in shaping the diffusion guidance.

We hypothesize that the influence of the condition at different timesteps represents the conflict between generation and guidance terms (Dinh et al., 2023). By addressing this, we can improve the well-known balance between diversity, fidelity, and condition reliance. Specifically, we introduce a progressive guidance mechanism where the values of $\omega$ are dependent on the timestep, and propose two solutions: heuristics-based guidance (cosine and linear) and computational, gradient-based guidance. We examine their impact on various tasks, including class-conditioned image, text-to-image, and text-to-motion generation. Our results corroborate our hypothesis (see Figure 1) and show that our proposed progressive weighting schemes consistently outperform the static baseline in terms of fidelity, diversity, and condition adherence. Since the proposed methods are low-cost and do not require training, they can be easily integrated into various diffusion guidance applications.

## 2 BACKGROUND

**Diffusion models** (Sohl-Dickstein et al., 2015; Ho et al., 2020; Song & Ermon, 2019) are a family of generative models that aim to convert noise into a target data distribution. Following DDPM (Ho et al., 2020), diffusion consists in training a network $\epsilon_\theta$ to denoise a noisy input to recover the original data at different noise levels, driven by a noise scheduler. The goal is to recover $x_0$, the original datapoint from $x_t = \sqrt{\gamma(t)}x_0 + \sqrt{1-\gamma(t)}\epsilon$, where $\gamma(t) \in [0, 1]$ is a monotonically decreasing noise scheduler function of the timestep $t$ and applied to a standard Gaussian noise $\epsilon \sim \mathcal{N}(0, 1)$. In practice, Ho et al. (2020) observed that predicting the added noise instead of $x_0$ yielded better performance. $\epsilon_\theta$ is then trained with the following loss based on the target data distribution $p_{\text{data}}$:

$$L_{\text{simple}} = \mathbb{E}_{x_0 \sim p_{\text{data}}, \epsilon \sim \mathcal{N}(0,1), t \sim \mathcal{U}[0,1]} \left[ \|\epsilon_\theta(x_t) - \epsilon\| \right] \quad . \tag{1}$$

Once the network is trained, we can sample from $p_{\text{data}}$ by setting $x_T = \epsilon \sim \mathcal{N}(0, 1)$ (with $\gamma(T)=0$), and gradually denoising to reach the data point $x_0 \sim p_{\text{data}}$ with different types of samplers e.g., DDPM (Ho et al., 2020) or DDIM (Song et al., 2020a).

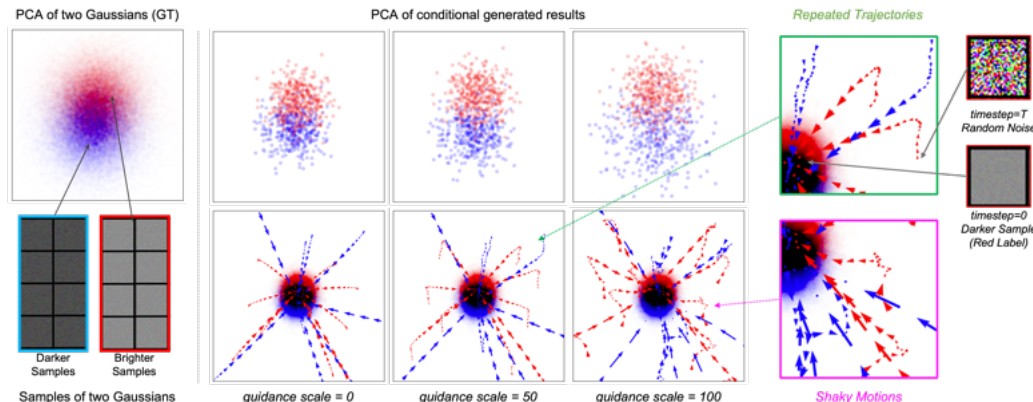

Figure 2: **Two-Gaussians Example.** We employ DDPM with CFG to fit two Gaussian distributions, a bright one (red) and a darker one (blue). The middle panel showcases samples of generation trajectories at different guidance scales $\omega$, using PCA visualization. Increasing guidance scale $\omega$ raises two issues: *repeated trajectory*: when $\omega=50$ the generation diverges from its expected direction before converging again, and *shaky motion*: when $\omega=100$ some trajectories wander aimlessly.

The diffusion task can also be seen from a score-matching perspective as the noise is proportional to the score of the perturbed data distribution (Song et al., 2020b) $\epsilon_\theta(x_t) \propto \nabla_{x_t} \log p(x_t)$. To leverage the condition $c$ and sample instead from $p(x_t|c)$, Dhariwal & Nichol (2021) use a pretrained classifier $p_\mu(c|x_t)$, forming $\nabla_{x_t} \log p(x_t|c) = \nabla_{x_t} \log p(x_t) + \nabla_{x_t} \log p(c|x_t)$ according to Bayes rule, with a scalar $\omega > 0$, which allows to control the amount of guidance added to the sampling. This yields the *classifier guidance* equation:

$$\hat{\epsilon}_\theta(x_t, c) = \epsilon_\theta(x_t) + (\omega + 1)\nabla_{x_t} \log p_\mu(c|x_t) \quad . \tag{2}$$

However, this requires training a classifier on noisy versions of the data, which can be cumbersome and impractical for novel classes. For this reason, with an implicit classifier from Bayes rule $\nabla_{x_t} \log p(c|x_t) = \nabla_{x_t} \log p(x_t, c) - \nabla_{x_t} \log p(x_t)$, Ho & Salimans (2022) propose to train a diffusion network on the joint distribution of data and condition by replacing $\epsilon_\theta(x_t)$ with $\epsilon_\theta(x_t, c)$ in Eq. 1. By dropping the condition during training, they employ a single network for both $\nabla_{x_t} \log p(x_t, c)$ and $\nabla_{x_t} \log p(x_t)$. This gives the *classifier-free guidance*, also controlled by $\omega$:

$$\hat{\epsilon}_\theta(x_t, c) = \epsilon_\theta(x_t, c) + \omega \left( \epsilon_\theta(x_t, c) - \epsilon_\theta(x_t) \right) \quad . \tag{3}$$

We can reformulate the above two equations into two terms: a *generation term* $\epsilon_\theta(x_t) \propto \nabla_{x_t} \log p(x_t)$ and a *guidance* term $\nabla_{x_t} \log p(c|x_t)$. The guidance term can be derived either from a pre-trained classifier or an implicit one, with $\omega$ balancing between generation and guidance.

## 3 GUIDANCE: PITFALLS AND CONCERNS

Here, we investigate the process of diffusion guidance. To this end, we first train a diffusion model on a synthetic dataset of $50,000$ images ($32 \times 32$) from two distinct Gaussian distributions: one sampled with low values of intensity (dark noisy images in the bottom-left of Figure 2), and the other with high-intensities (bright noisy images). The left-top of Figure 2 shows the PCA (Kambhatla & Leen, 1997)-visualised distribution of the two sets, and the left-bottom part shows some ground-truth images. To fit these two labeled distributions, we employ DDPM (Ho et al., 2020) with *classifier-free guidance* (Ho & Salimans, 2022) conditioned on intensity labels.

Upon completion of the training, we can adjust the guidance scale $\omega$ to balance between the sample fidelity and condition adherence, illustrated in the right part of Figure 2. The first row depicts the variations in generated distributions on different $\omega$ (from 0 to 100), visualized by the same PCA parameters. The second row shows the entire diffusion trajectory for sampled data points (same seeds across different $\omega$): progressing from a random sample (*i.e.*, standard Gaussian) when $t = T$ to the generated data (blue or red in Figure 2) when $t = 0$.

**Emerging issues and explainable factors.** As the guidance scale increases, the resulting distributions of the two labels increasingly diverge. This is expected due to the increasing *guidance term*

in Eq. 3 dragging the generation away from the different labels. This separation often comes at the cost of fidelity, leading to distortions in the shape of the generated results (see Figure 2 first row). Additionally, as depicted in the second row of Figure 2, two main issues arise here: (i) *repeated trajectories* that initially diverge from their expected convergence path before redirecting towards the correct direction (see Figure 2 when $w = 50$); and (ii) highly shaky motions that wander aimlessly along the trajectory, which is more pronounced at scale 100.

These two issues can be attributed to two possible factors: (1) incorrect classification prediction, and (2) an initial gradient direction received from the *guidance term* that may conflict with the *generation term*, *i.e.*, the force that brings the process towards the two Gaussian distributions.

Addressing the former factor (optimal guidance generation) requires a *flawless* classifier, whether explicit as in *guided diffusion*, or implicit as in *classifier-free guidance*. However, discerning between two highly noisy data, e.g. when the timestep approaches $T$, is a very challenging classification task unlikely to perform perfectly. Consequently, incorrect classification will inevitably steer the generation towards the wrong direction, thus generating shaky trajectories (see $\omega = 100$ in Figure 2). A similar observation is reported by Li et al. (2023), where the implicit classifier within the *classifier-free guidance* diffusion framework is analyzed. One of their findings reveals that relying solely on single steps for label prediction yields low accuracy at larger timesteps (i.e., noisier images).

Regarding the latter factor (conflicting gradients), Figure 2 provides a visual clue by comparing the scale between 50 and 0. The trajectories exhibit *translational moves* at the initial stage due to the strong force of the classifier, which wishes to increase the distance between different classes, as a discriminative model does. Consequently, this trajectory inversely progresses in a U-turn before gravitating towards the convergence region (repeated trajectory in Figure 2). We argue that these anomalies arise from the conflict between the *guidance term* and the *generation term* in Eq. 3. This is in line with (Dinh et al., 2023), which also reports similar gradient direction conflicts within the classifier guidance paradigm between the classification term and other terms.

Both issues point to the same flaw in diffusion guidance: in the initial stages of the generation, the *guidance term* does not necessarily steer the generation optimally, and may even impede generation performance. Our hypothesis is that such **erratic behavior** could be a contributing factor to the well-known performance **dichotomy between fidelity and condition adherence** for diffusion guidance (Ho & Salimans, 2022; Dhariwal & Nichol, 2021). Specifically, misdirected trajectories might culminate in regions of the uncharted area during training, thereby affecting generation performance.

## 4 PROGRESSIVE WEIGHTING SCHEMES

In this work, we introduce Progressive-Guidance, a simple yet efficient method to address the two aforementioned issues during guidance, where the generation may suffer a longer and noisier trajectory. A direct and intuitive remedy is to adopt a progressive guidance scale $\omega(t)$, as opposed to the original static one $\omega$ (Ho & Salimans, 2022; Dhariwal & Nichol, 2021). This progressive scale is moderated during generation, when $t \to T$, and progressively amplified as $t$ approaches 0. Therefore, the generation process of *classifier-free guidance* becomes: $\hat{\epsilon}_\theta(x_t, c) = \epsilon_\theta(x_t, c) + \omega(t)(\epsilon_\theta(x_t, c) - \epsilon_\theta(x_t))$. To achieve $\omega(t)$, we propose two strategies: a heuristic one (Section 4.1) and a computational gradient-based one (Section 4.2).

### 4.1 HEURISTIC APPROACH

Our goal is to design a guidance weighting scheme that attenuates the influence of guidance in the initial stages to avoid repeated trajectories and shaky motions, and amplifies it in the later stages when the guidance term is more trustworthy. To this end, we get inspired by the parameterized power-cosine-like curve introduced in (Gao et al., 2023)[1] and instead propose a more generic, simple and systematic form. Specifically, we introduce two heuristic functions for progressive weighting: *linear* $\omega(t){=}(1{-}t/T)$ and *cosine* $\omega(t){=}cos(\pi t/T){+}1$, where $T$ is the maximum diffusion timestep.

To integrate $\omega(t)$ into the generation at a given static guidance scale value, we standardize $\omega(t)$ to conserve the overall energy of the progressive weights, making it equal to the static scale $\omega$ in Eq. 3:

---

[1]For comparison against (Gao et al., 2023) see Appendix A.7.

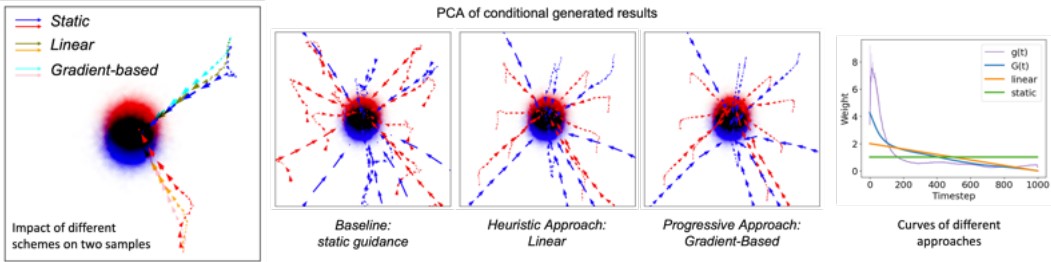

Figure 3: **Proof of Concept.** Our proposed *linear* and *gradient-based* progressive weighting scheme effectively minimize the aforementioned issues during the generation and reduce the repeated trajectories and temper the shaky motion. Zoomed comparison at left panel shows two single generation trajectories. The progressive curves are illustrated at right panel.

$\int_0^T \omega(t)dt = \omega T$. This standardization ensures that static and progressive guidance scales yield similar condition controls during generation, and lower the influence from the shape of the curve. With the linear function, the standardized guidance scale is: $\omega(t) = 2(1 - t/T)\omega$, where $\omega$ retains similar significance as in the static guidance; with cosine, it remains the same as $\int_0^T \cos(\pi t/T) + 1 dt = T$.

## 4.2 COMPUTATIONAL APPROACH

Computing the optimal weighting curve could be cast as an optimization problem, seeking for the best curve shape that maximizes metrics such as FID or Inception Score. This, however, is intractable because it would require simultaneous optimization over all steps and for a large set of images. Alternatively, we introduce a novel perspective that exploits gradient information. We discover that the gradient response of the condition offers insights for crafting an efficient progressive weighting curve. This gradient-driven method demands less computation than full optimization; yet it excels over heuristic approaches and adapts to varying architectures and datasets.

**Gradient norm on condition.** Let us look at the norm of the gradient from Eq. 2:

$$\nabla_c \hat{\epsilon}_\theta (x_t, c) = \nabla_c \nabla_{x_t} \log p(c|x_t) = \nabla_c \epsilon_\theta (x_t, c). \tag{4}$$

The norm of this gradient $g(t, c) = \|\nabla_c \epsilon_\theta (x_t, c)\|$ illustrates the *influence* of the condition $c$ on the generation of $\epsilon(x_t, c)$ at timestep $t$. If the value of $\epsilon(x_t, c)$ remains consistent across $c$ values (*i.e.*, the gradient is zero), it might indicate that the condition does not significantly influences the generation process at this timestep, thus implying a less reliable classifier $p(c \mid x_t)$. Importantly, $g(t, c)$ is readily derivable from the L1 training loss by its expectation:

$$g(t, c) = \mathbb{E}_t[\|\nabla_c \|\epsilon(x_t, c) - \epsilon\|_1\|] \quad . \tag{5}$$

**Cumulative gradient norm.** Given the sequential nature of the diffusion generation process (Ho et al., 2020), we opt to *accumulate* the independently measured presence $g(t)$ of each condition's information from the beginning to the end, resulting in its cumulative function $G(t, c) = \sum_{i=t}^T g(i, c)$. From this viewpoint, the *linear* heuristic curve can be interpreted as a naive assumption that the condition is *uniformly influential* across all timesteps, and the *cosine* corresponds to a sine shape curve centered at the mid-point of the timestep. We hence leverage our *gradient-based computational* progressive weighting $G(t, c)$ as a direct substitute for $\omega(t)$. Unlike the heuristic functions, this progressive weighting not only improves the fidelity and diversity (Section 5) but also eases applicability across diverse diffusion methods, datasets, and even schedulers. In practice, $G(t, c)$ can be measured by the gradient norm of the condition variable w.r.t. the L1 loss during a training-like stage, named *probing*, which can be executed during or post-training in only a few epochs.

**Class-dependent weighting.** During probing, we can compute the *class-dependent* weighting $G(t, c)$, recording the gradnorm with class label or the *class-independent* weighting by averaging classes $G(t) = \sum_c (G(t, c))/C$ where $C$ is the number of labels. Note that the class-dependent one may not be tractable for some tasks (*e.g.*, text-to-image generation), due to the embedding form of the condition being not countable, or in datasets such as CIN-256 which contain 1000 labels, where probing the gradient on each label would require prohibitively expensive computational time.

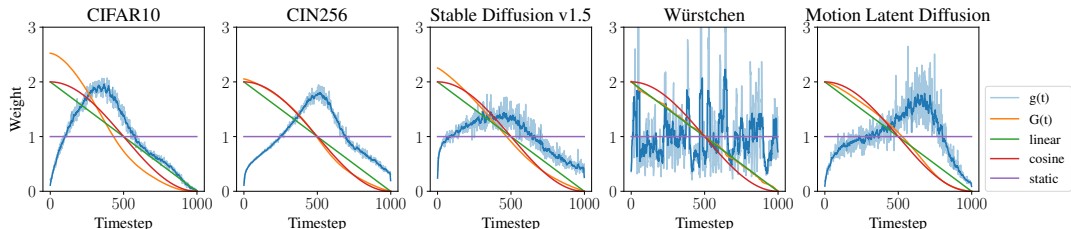

Figure 4: Standardized gradient on condition $g(t)$ and its cumulative function $G(t)$ curves computed from CIFAR10-DDPM (Ho et al., 2020), CIN256-LDM (Rombach et al., 2022) and SDv1.5 (Rombach et al., 2022), Wuerstchen (Pernias et al., 2023), and MLD (Chen et al., 2023) architectures separately (before and after smoothing), compared with linear, cosine and static weighting schemes.

**Proof of concept.** We assess our progressive methods using the two-Gaussians example in Figure 3. Relative to static weighting, the *linear* and *gradient-based* approaches notably diminish repeated trajectories and stabilize the motion. The left image in Figure 3 compares two deterministic samples from each label, with the gradient-based method producing the smoothest trajectories.

On the right panel, we show the standardized gradient-based curve $G(t)$ compared to linear and static ones. We clearly observe that the proposed gradient-based method focuses more on the later stage. Additionally, we evaluate all methods on several datasets and tasks (see Section 5 for dataset details) and illustrate their standardized gradient on $g(t)$ in Figure 4. We observe that the influence of the condition is weak at the initial stage ($t \to T$) due to the high level of noise, and then it gradually increases, peaking around the intermediary stages. Notably, the peak varies for different datasets and architectures. When $t \to 0$, the influence of the condition reduces as the data's principal information has been reconstructed. The final denoising steps offer refinement and clean-up, as also observed in P2-Weighting (Choi et al., 2022) and min-SNR (Hang et al., 2023). Ablations on all progressive weighting, including a negative perturbation experiment, can be found in Appendix A.1.

## 5 EXPERIMENTS

We experiment on three tasks: (i) class-conditioned generation (Section 5.1); (ii) text-conditional image generation (Section 5.2); and text-to-motion generation (Section 5.3). (Ablation in Appendix A.1). We compare the *baseline (static)* weighting scheme against the proposed Progressive-Guidance methods: heuristics *linear* and *cosine*, and gradient-based *gn* and *gn-all*. Specifically, *gn* averages the gradient norm of all labels to create G(t), while *gn-all* is label-dependent (G(t,c)), i.e. the weighting is applied separately for each label. We illustrate each method with a color: blue for *baseline*, orange for *linear*, green for *cosine*, red for *gn* and purple for *gn-all*.

### 5.1 CLASS-CONDITIONAL IMAGE GENERATION

**Datasets and Metrics.** Two datasets are involved: *CIFAR-10 (Krizhevsky, 2009)* comprises $60,000$ images, each of resolution $32 \times 32$, distributed across 10 classes; *ImageNet (CIN)-256 (Deng et al., 2009)* comprises 1.2M images of resolution $256 \times 256$ with $1,000$ labels.
For evaluation, we compute the FID (Fréchet inception distance) vs. IS (Inception Score) for a generated sample set of $50,000$ images, contrasting against the validation set of each dataset.

**Models.** For CIFAR-10, we employ the DDPM (Ho et al., 2020) framework, which denoises directly in the image domain. Inference employs the DDIM (Song et al., 2020a) framework with 200 timesteps. For CIN-256, we use the Latent Diffusion Model (LDM) (Rombach et al., 2022), where diffusion is conducted within a pre-trained VAE space. Inference leverages a 50-step DDIM process with a public checkpoint of official implementation reporting FID of 3.6[2].

**Results and analysis.** Figure 5a and Figure 5c depicts the FID vs. IS curves for both datasets (detailed results in Tables 2-3 in Appendix). We make the following observations: (1) All progressive

---

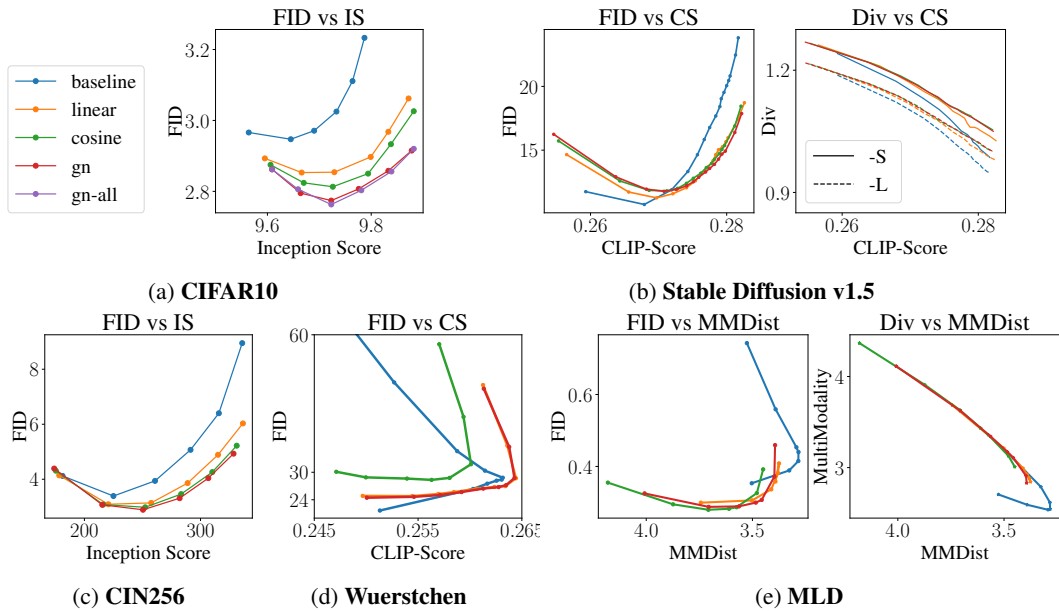

Figure 5: **Trade-offs** observed in (a) CIFAR10, (b) SDv1.5, (c) CIN256, (d) Wuerstchen, and (e) Motion Latent Diffusion (MLD). We observe that our progressive weighting schemes (*linear*, *cosine*, and gradient-based methods *gn* and *gn-all*) consistently outperform the baseline static weighting scheme (blue) across all experimental setups in terms of FID, especially when the guidance scale is strong. In terms of diversity, as demonstrated in (b) with SD-v1.5, all progressive methods showcase enhanced diversity corresponding to a particular guidance level (as indicated by the CLIP-Score).

methods outperform the baseline on both codebases (DDPM or LDM) and datasets. (2) Among progressive methods, gradient-based methods result in the best performance, followed by the *cosine* and then the *linear* one. (3) The gain from using information from *all* labels (i.e., *gn-all*) is relatively marginal against averaged *gn* (see Table 2, FID 2.774 vs. 2.764). (4) Due to our standardization process (see Section 4.2), the optimal guidance scale remains close to the baseline across different shapes of progressive weighting curve, indicating a similar semantic meaning to the static one.

## 5.2 TEXT-TO-IMAGE GENERATION

**Dataset and Metrics.** The training dataset is LAION (Schuhmann et al., 2022), which contains 5B high-quality images with paired textual descriptions. For evaluation, we use the smaller COCO (Lin et al., 2014) validation dataset, which contains 30, 000 text-image paired data.

We use three metrics: (i) *FID* to examine the fidelity of generated images; (ii) *CLIP-Score (CS)* (Radford et al., 2021) to assess the alignments between the generated images and their corresponding text prompts; (iii) *Diversity (Div)* to measure the model's capacity to yield varied content. For this, we compute the standard deviation of image embeddings via Dino-v2 (Oquab et al., 2023) from multiple generations of the same prompt (Discussion for DINO-v2, CLIP embeddings in Appendix A.3). We compute FID and CS for a sample set of 10, 000 images against the COCO dataset in a zero-shot fashion (Rombach et al., 2022; Saharia et al., 2022). For diversity, we resort to two text description subsets from COCO: one with the 1000 *longest captions* and another with the 1000 *shortest captions* (labelled as -L and -S in Figure 5b). These represent varying descriptiveness levels; longer captions inherently provide more specific conditions than shorter ones, presumably leading to less diversity. We produce 10 images for each prompt using varied sampling noise.

**Model.** We use the following models: (1) Stable Diffusion (SD) (Rombach et al., 2022), which uses the CLIP (Radford et al., 2021) text encoder to transform textual inputs into embeddings. We use the public checkpoint of SD v1.5 [3] and employ DDIM sampler with *50* steps. (2) Wuerstchen (Pernias

---

[3]https://huggingface.co/runwayml/stable-diffusion-v1-5

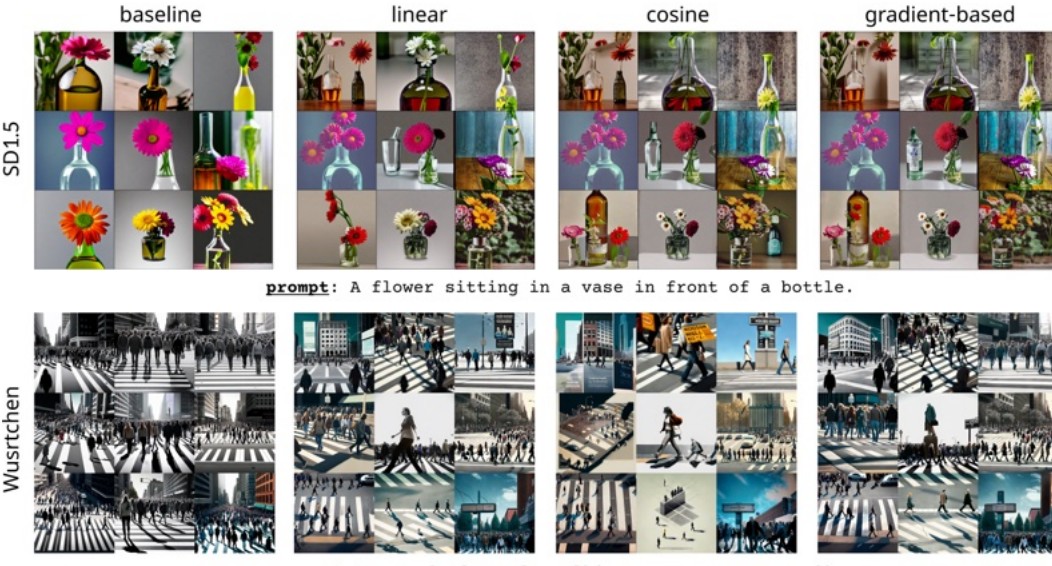

Figure 6: **Qualitative Results.** Comparison of generated results with SD and Wuerstchen.

et al., 2023), which trains text-conditional models by compressing the text-conditional stage into a latent space, achieving comparable results to SD with cheaper computational cost [4].

**Results and analysis.** We display the FID vs. CS curves in Figure 5b for Stable Diffusion, and Figure 5d for Wuerstchen (see Appendix A.4 for a full detailed table). Here, we expect an optimal balance between a high CS and a low FID (right-down corner of the graph). Left panel in Figure 5b shows the results of SD, and we can see that all proposed methods (heuristics and gradient-based) perform overall similarly, and lead to better performance than the baseline. However, gradient-based *gn* presents marginally superior results than the *linear* and *cosine* heuristics, whereas *linear* recorded lower FID than the rest. The baseline regresses FID fast when CS is high, but generates the best FID when CS is low, i.e., low condition level. Figure 5d shows results for Wuerstchen. We observe that with progressive guidance, both *gradient-based* and *linear* methods can reach higher CLIP scores on Wuerstchen than the baseline static guidance, while not compromising too much on FID. This is not the case for the baseline method, which retraces CLIP score and results in much worse FID.

The results of Div vs. CS on different guidance scales for Stable Diffusion are illustrated in the right panel of Figure 5b. Our findings reveal that: (i) as hypothesized, longer captions indeed exhibit reduced diversity compared to shorter ones; (ii) for a given CS, the gradient-based and cosine methods consistently lead to superior diversity than the linear one. Finally, all proposed Progressive-Guidance methods outperform the baseline across both subsets and different guidance scales.

**Qualitative results.** Figure 6 depicts two sets of generations from SDv1.5 and Wuerstchen with $\omega=10$ for all images. We observe that progressive weighting enhances the diversity of outputs in terms of composition, color palette, style and image quality by refining shades and enriching textures. Notably, some baseline samples overlook textual cues, e.g., in front of a bottle for SD1.5, while our methods capture these nuances correctly. We observe that the heuristic and gradient-based methods perform visually similarly, where the latter generates more photorealistic images with emphasis on fine-grained details, such as the shading of crowds, the background of the vase and the petals of flowers. (More results in Appendix A.5)

**User study.** To perceptually verify the effectiveness of our methods, we performed a user study. We present users with a pair of mosaics of 9 generated images each and ask them to vote for the best in terms of realism, diversity and text-image alignment. Each pair compares baseline generations against either linear, cosine or gradient-based ones. For Stable Diffusion, as shown in Figure 9, our results reveal that over $60\%$ of users consider our generated images more realistic and better aligned

---

[4]In our experiments, we use the public checkpoint `https://github.com/dome272/Wuerstchen`

with the text prompt, while approximately $80\%$ find our generations more diverse. Similarly for the *Wuerstchen* model (Figure 10), more than $60\%$ of users favored the realism and congruence with the text of our generated images. Nevertheless, an equal proportion of users expressed a preference for the text alignment presented by the baseline model. We postulate that this divergence in opinion may arise from our results being overly diverse and inconsistent, potentially obscuring the inherent semantic nuances embedded within the images. This corroborates our findings that static weighting is perceptually inferior to progressive weightings. More details in Appendix A.4.1.

### 5.3 TEXT-TO-MOTION GENERATION

**Dataset and Metrics.** We use data from the widely-adopted HumanML3D (Guo et al., 2022), which assigns 44,970 textual descriptions to a subset of 14,616 motions from the AMASS (Mahmood et al., 2019) dataset. We use three metrics: (i) FID to measure the fidelity of generations; (ii) Multi-Modal Distance (MMDist) to assess the alignment between the text and motion; (iii) MultiModality to assess the ability to generate varied motions under the same conditioning signal. The goal is to achieve low FID and MMdist, and high MultiModality.

**Model.** We use the diffusion-based Motion-Latent-Diffusion model (MLD) [5] trained with CFG as the base model. MLD operates on the latent space of a motion VAE.

**Results and analysis.** Similar to Section 5.1 and following generative animation works (Zhang et al., 2023; Tevet et al., 2023), we compare in Figure 5e MMDist with FID and generation diversity (MultiModality in animation). W.r.t. MMDist vs. FID, we observe that the proposed methods outperform the baseline by reaching closer to the down-right optimal region. Notably, the baseline obtains the lowest value for MMDist, yet sacrifices on FID. We also note that the cosine heuristic marginally outperforms all proposed methods. For MMDist vs. MultiModality, all approaches perform similarly, probably because of the dataset's small size and limited motion diversity relative to image variety. The similarities in diversity across all methods can be attributed to the small size and limited variation inherent in the animation dataset. Overall, our findings suggest that our progressive weighting methods enhance the balance between FID and MMDist without requiring retraining.

## 6 DISCUSSION

In this work, we delved into the effect of *classifier-free guidance* throughout the denoising process and found that using a constant guidance weight leads to conflicting trajectories. To solve this, we proposed general, low-cost and effective progressive weighting schemes to mitigate the conflict among generation quality, condition adherence, and diversity. We showed that the heuristic weighting schemes (*cosine* and *linear*) result in improved overall performance for fidelity, image quality and diversity compared to a static weighting scheme. These heuristics serve as *ready-to-integrate* modules complementing the diffusion generation process for a wide range of diffusion models and are directly applicable even after a model is trained. Additionally, we introduced a computational approach based on the condition gradient norm $g(t)$ and showed that it also leads to performance gains. This indicates that $g(t)$ provides informative insights on the reliance on the condition during denoising timestep $t$. Thus, we hypothesize that it can serve as a tool for downstream tasks of conditional DDPMs (see Textual Inversion in Appendix A.2).

**Limitations.** Throughout our experiments, we observed two artifacts of applying progressive weighting: (i) overly-contrasted images, due to overshooting weights at the final stage of generation (where the texture is formed); and (ii) excessive muting during the initial stage which can sometimes disrupt structural integrity, particularly salient in areas like human or animal faces. However, given that our method is low-cost and does not require any re-training, the progressive curve can be fine-tuned for a specific content to achieve a balance between these artifacts and diversity. Finally, we acknowledge the absence of a formal proof of optimality of the gradient-based curves; we leave this as direction for future work.

---

[5] Public checkpoint at https://github.com/ChenFengYe/motion-latent-diffusion

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

# A APPENDIX

In the appendix, we demonstrate ablation studies, more experiments, some detailed results complementary to the main paper, and more qualitative demonstrations.

## A.1 ABLATIONS

In this section, we show some ablations of our proposed methods on the following aspects: DDIM steps; different heuristic curve shape. We also report a negative perturbation test.

**Different curves.** We compare multiple primitive curves on the CIFAR10-DDPM setup for FID and IS metrics including *cosine, sine* and *linear with inverted linear* on timestep with different guidance scales. The results are showed in Figure 7a. We notice that all the increasing-trended $(T : 1000 \rightarrow 0)$ curves perform better than the baseline (static), which is significantly better than the decreasing-trended curves. The observation adheres to our principal idea of compressing the weights at the initial stage of the diffusion process.

**Different DDIM steps.** DDIM sampler allows for accelerated sampling (e.g., 50 steps as opposed to 1000) with only a marginal compromise in generation performance. In this ablation study, we evaluate the effectiveness of our progressive weighting schemes across different sampling steps. We use the CIN256-LDM codebase, with the same configuration as our prior experiments in Section 5.1. We conduct tests with $50, 100,$ and $200$ steps, for baseline, two heuristics (linear and cosine) and our computational gradient-based method, all operating at their optimal guidance scale in Tab 3. The results, FID vs. IS for each sampling step, are presented in Tab. 1. We observe that the performance of heuristic and gradient-based progressive weighting schemes remains stable across different timesteps with the gradient-based method achieving the best FID vs. IS performance.

**Negative Perturbation.** In this ablation study, we analyse the importance of different intervals on timesteps by *independently* setting the guidance scale to zero over different intervals, each spanning $50$ timesteps, while retaining the default guidance scale of $0.15$ in other intervals throughout the entire generation process, resulting in 20 independent intervals. For this, we use the CIFAR-10-DDPM dataset and adopt the same codebase as in previous experiments.

By recording the FID as the results of each interval perturbed weighting scheme, we aim to measure the importance and contribution of each interval on the overall generation quality. As illustrated in Figure 7b, perturbing the intervals at initial stages, such as $[800, 850)$ or $[850, 900)$, yields improved FID metrics compared to other intervals. The most significant FID regression is observed just before the ending stages, with a general increase trend as we move from $T = 1000$ to $T = 0$. This observation aligns with our main argument that classifier issues mainly arise at the initial stages, and a progressively increasing weighting scheme can ameliorate the performance of diffusion guidance.

## A.2 TEXTUAL INVERSION

Prompted by the intuition that the conditional information is not uniformly important across timesteps, we further explore our findings in akin applications, such as the task of textual inversion (Gal et al., 2022). Given a few example images, the goal is to learn a new word which can be incorporated in the text condition and guide the generation towards novel instances that retain characteristics of the examples. The word is learnt using a reconstruction objective, i.e., given a random diffusion step and the condition signal the goal is to produce images that are visually similar to the examples by optimizing the condition embedding. In the original formulation, the diffusion timesteps are *uniformly* sampled across the timestep scale, e.g., from 0 to 1000.

Table 1: **Ablation on sampling steps DDIM.** Experiment on CIN-256 and Latent Diffusion Model

| steps | baseline (static) FID↓ | IS↑ | linear FID↓ | IS↑ | cosine FID↓ | IS↑ | gn FID↓ | IS↑ |
|---|---|---|---|---|---|---|---|---|
| 50 | 3.393 | 220.6 | 3.090 | 225.0 | 2.985 | 252.4 | 2.888 | 250.5 |
| 100 | **3.216** | 229.8 | **2.817** | 225.2 | 2.818 | 255.3 | 2.750 | 252.6 |
| 200 | 3.222 | 229.5 | 2.791 | 223.2 | **2.801** | 254.3 | **2.714** | 251.3 |

Table 2: **Experiment on CIFAR-10 DDPM.** We evaluate the FID and IS results for the baseline, two heuristic methods (linear and cosine), and the gradient-based methods, both with and without condition-dependent information. Best FID and IS are highlighted.

| guidance | baseline (static) | | linear | | cos | | gn | | gn-*all* | |
|---|---|---|---|---|---|---|---|---|---|---|
| | FID↓ | IS↑ | FID↓ | IS↑ | FID↓ | IS↑ | FID↓ | IS↑ | FID↓ | IS↑ |
| 0.10 | 2.966 | 9.564 | 2.893 | 9.595 | 2.875 | 9.606 | 2.862 | 9.609 | 2.862 | 9.609 |
| 0.15 | **2.947** | 9.645 | **2.853** | 9.666 | 2.824 | 9.670 | 2.795 | 9.664 | 2.806 | 9.659 |
| 0.20 | 2.971 | 9.690 | 2.854 | 9.729 | **2.813** | 9.726 | **2.774** | 9.723 | **2.763** | 9.723 |
| 0.25 | 3.025 | 9.733 | 2.897 | 9.799 | 2.850 | 9.794 | 2.807 | 9.776 | 2.803 | 9.781 |
| 0.30 | 3.111 | 9.764 | 2.968 | 9.833 | 2.933 | 9.838 | 2.858 | 9.833 | 2.856 | 9.839 |
| 0.35 | 3.233 | **9.787** | 3.062 | **9.872** | 3.026 | **9.882** | 2.915 | **9.878** | 2.920 | **9.882** |

Our observations suggest that the influence exerted by the given condition varies across different timesteps. To further demonstrate this argument, we conducted a qualitative experiment, selectively optimizing the condition embedding across specific timestep ranges. As depicted by the model's gradient response $g(t)$ in Figure 11, conducting textual inversion at distinct timesteps produces vastly disparate results. Specifically, when sampling from *low influence* regions, for instance, *t=(0,100)* or *t=(900,1000)* (Figure 11, red and purple area), the generated images show low resemblance to the ground truth reference. Conversely, sampling around *high influence* intervals, i.e., $t = (350, 450)$, yields images high similarity to the ground truth, akin to those generated by sampling across all timesteps ($t = (0, 1000)$). More interestingly, by focusing on a *singular* timestep, $t = 400$, which represents the $g(t)$ peak and thus indicates maximal influence (Figure 11, orange line at top), the derived image captures the essence of the ground truth and stands in contrast to those generated from *low influence* regions.

## A.3  DISCUSSION ON DIVERSITY

Diversity plays a pivotal role in textual-based generation tasks. Given similar text-image matching levels (usually indicated by CLIP-score), higher diversity gives users more choices of generated content. Most applications require higher diversity to prevent the undesirable phenomenon of content collapsing, where multiple samplings of the same prompt yield nearly identical or very similar results. We utilize the standard deviation within the image embedding space as a measure of diversity. This metric can be derived using models such as Dino-v2 (Oquab et al., 2023) or CLIP (Radford et al., 2021). Figure 8 provides a side-by-side comparison of diversities computed using both Dino-v2 and CLIP, numerical results are also reported in Table. 5. It is evident that Dino-v2 yields more discriminative results compared to the CLIP embedding. While both embeddings exhibit similar trends, we notice that CLIP occasionally produces a narrower gap between long captions (-L) and short captions (-S). In some instances, as depicted in Figure 8, CLIP even reverses the order, an observation not apparent with the Dino-v2 model. However, our methods are consistently outperforming the baseline on both metrics.

## A.4  DETAILED TABLE OF EXPERIMENTS

In this section, we show detailed table of experiments: CIFAR-10-DDPM (Table 2), CIN256-LDM (Table 3), Wuerstchen (Table 4) and Stable Diffusion (Table 5)

### A.4.1  USER STUDY

In this section, we elaborate on the specifics of our user study setup and present the corresponding results.

For the evaluation, each participant was presented with a total of 10 image sets. Each set comprised 9 images. Within each set, three pairwise comparisons were made: baseline vs. gradient-based, baseline vs. linear, and baseline vs. cosine. Throughout the study, two distinct image sets (equivalent to 20 images for each method) were utilized. We carried out two tests for results generated with stable diffusion and Wuerstchen respectively.

Table 3: **Experiment on CIN-256 LDM.**, we present similar metrics to Tab. 2. The FID vs. IS values for the baseline, heuristic linear and cosine, and gradient-based method *gn* are showed. We notice a consistent observation: the gradient-based*gn* method outperforms other in terms of FID-IS balance, with *cosine* trails closely behind. The linear heuristic performs worse than previous two, but all progressive methods are far superior to the baseline.

| guidance $w$ | baseline: *static* | | prog: *linear* | | prog: *cosine* | | prog: *gn* | |
|---|---|---|---|---|---|---|---|---|
| | FID↓ | IS↑ | FID↓ | IS↑ | FID↓ | IS↑ | FID↓ | IS↑ |
| 0.4 | 4.117 | 181.2 | 4.136 | 178.3 | 4.311 | 175.4 | 4.394 | 174.0 |
| 0.6 | **3.393** | 225.0 | **3.090** | 220.6 | 3.083 | 216.2 | 3.068 | 215.4 |
| 0.8 | 3.940 | 260.8 | 3.143 | 257.5 | **2.985** | 252.4 | **2.888** | 250.5 |
| 1.0 | 5.072 | 291.4 | 3.858 | 288.9 | 3.459 | 283.3 | 3.308 | 282.0 |
| 1.2 | 6.404 | 315.8 | 4.888 | 315.1 | 4.256 | 310.1 | 4.041 | 306.7 |
| 1.4 | 8.950 | **335.9** | 6.032 | **336.5** | 5.215 | **331.2** | 4.934 | **328.4** |

Table 4: **Experiment on Wuerstchen.**, The FID vs. CLIPScore(CS) values for the baseline, heuristic linear and cosine, and gradient-based method *gn* are showed.

| guidance $w$ | baseline: *static* | | prog: *linear* | | prog: *cosine* | | prog: *gn* | |
|---|---|---|---|---|---|---|---|---|
| | FID↓ | CS↑ | FID↓ | CS↑ | FID↓ | CS↑ | FID↓ | CS↑ |
| 2 | 21.7 | 0.251 | - | - | - | - | - | - |
| 5 | 26.4 | 0.260 | 24.8 | 0.250 | - | - | 24.4 | 0.250 |
| 7 | 27.5 | 0.261 | 24.8 | 0.254 | - | - | 24.64 | 0.254 |
| 9 | 28.2 | 0.263 | 25.3 | 0.257 | 30.1 | 0.247 | 25.0 | 0.257 |
| 11 | 28.4 | 0.263 | 25.7 | 0.258 | 28.9 | 0.250 | 26.4 | 0.261 |
| 15 | 28.8 | 0.263 | 26.4 | 0.261 | 28.6 | 0.253 | 26.4 | 0.261 |
| 20 | 30.4 | 0.261 | 26.8 | 0.262 | 28.3 | 0.256 | 26.8 | 0.263 |
| 25 | 34.6 | 0.259 | 27.0 | 0.263 | 28.8 | 0.258 | 27.2 | 0.26 |
| 40 | 49.6 | 0.253 | 28.7 | 0.264 | 31.8 | 0.260 | 28.7 | 0.264 |
| 60 | 62.3 | 0.248 | 35.8 | 0.264 | 42.1 | 0.259 | 35.5 | 0.264 |
| 80 | 69.8 | 0.244 | 49.0 | 0.261 | 57.9 | 0.257 | 48.22 | 0.261 |

Table 5: **Experiment on Stable Diffusion v1.5.**, we present FID vs. CLIP-Score (CS) and diveristy generated from Dino-v2 and CLIP embeddings separately. Including methods of the baseline, heuristic linear and cosine, and gradient-based methods are showed. We notice that the *gradient-based* method outperforms other in terms of FID-CS and diversity balance, with *cosine* reaching good diversity but slightly worst FID vs. CS. The linear heuristic performs worse than previous two, but all progressive methods are far superior to the baseline when CS is high. The baseline keeps the best FID of 10.741 at guidance 3 but regresses rapidly when CLIP-Score is bigger.

| | $w$ | 1 | 2 | 3 | 4 | 5 | 6 | 7 | 8 | 9 | 10 | 11 | 12 | 13 | 14 | 20 | 25 |
|---|---|---|---|---|---|---|---|---|---|---|---|---|---|---|---|---|---|
| **baseline** | clip-score | 0.2593 | 0.2679 | 0.2719 | 0.2743 | 0.2757 | 0.2767 | 0.2775 | 0.2784 | 0.2790 | 0.2792 | 0.2796 | 0.2800 | 0.2803 | 0.2805 | 0.2813 | 0.2817 |
| | FID | 11.745 | 10.741 | 11.887 | 13.328 | 14.639 | 15.832 | 16.777 | 17.682 | 18.419 | 19.031 | 19.528 | 20.058 | 20.462 | 20.818 | 22.463 | 23.810 |
| | Div-CLIP-L | 0.315 | 0.300 | 0.289 | 0.281 | 0.275 | 0.271 | 0.267 | 0.263 | 0.260 | 0.259 | 0.257 | 0.255 | 0.254 | 0.253 | 0.250 | 0.251 |
| | Div-Dinov2-L | 1.188 | 1.122 | 1.083 | 1.053 | 1.033 | 1.018 | 1.007 | 0.996 | 0.987 | 0.982 | 0.976 | 0.971 | 0.967 | 0.962 | 0.951 | 0.948 |
| | Div-CLIP-S | 0.317 | 0.300 | 0.288 | 0.280 | 0.273 | 0.268 | 0.263 | 0.260 | 0.256 | 0.254 | 0.252 | 0.251 | 0.249 | 0.248 | 0.246 | 0.246 |
| | Div-Dinov2-S | 1.241 | 1.173 | 1.131 | 1.103 | 1.082 | 1.065 | 1.051 | 1.042 | 1.031 | 1.024 | 1.019 | 1.013 | 1.006 | 1.003 | 0.992 | 0.986 |
| **linear** | clip-score | 0.2565 | 0.2656 | 0.2697 | 0.2721 | 0.2741 | 0.2754 | 0.2763 | 0.2772 | 0.2780 | 0.2784 | 0.2788 | 0.2794 | 0.2799 | 0.2802 | 0.2817 | 0.2826 |
| | FID | 14.649 | 11.718 | 11.260 | 11.581 | 12.056 | 12.596 | 13.147 | 13.670 | 14.179 | 14.651 | 15.032 | 15.270 | 15.663 | 15.969 | 17.478 | 18.718 |
| | Div-CLIP-L | 0.320 | 0.308 | 0.300 | 0.294 | 0.289 | 0.285 | 0.281 | 0.278 | 0.275 | 0.273 | 0.271 | 0.270 | 0.268 | 0.267 | 0.262 | 0.259 |
| | Div-Dinov2-L | 1.209 | 1.153 | 1.119 | 1.094 | 1.076 | 1.060 | 1.048 | 1.039 | 1.030 | 1.024 | 1.016 | 1.011 | 1.006 | 1.002 | 0.986 | 0.979 |
| | Div-CLIP-S | 0.324 | 0.311 | 0.302 | 0.296 | 0.291 | 0.287 | 0.282 | 0.280 | 0.277 | 0.273 | 0.271 | 0.271 | 0.270 | 0.269 | 0.263 | 0.261 |
| | Div-Dinov2-S | 1.262 | 1.210 | 1.172 | 1.147 | 1.129 | 1.113 | 1.099 | 1.091 | 1.082 | 1.069 | 1.060 | 1.060 | 1.057 | 1.053 | 1.038 | 1.027 |
| **cosine** | clip-score | 0.2553 | 0.2643 | 0.2686 | 0.2712 | 0.2728 | 0.2741 | 0.2751 | 0.2762 | 0.2770 | 0.2778 | 0.2782 | 0.2789 | 0.2793 | 0.2797 | 0.2812 | 0.2821 |
| | FID | 15.725 | 12.587 | 11.846 | 11.810 | 12.009 | 12.400 | 12.796 | 13.197 | 13.629 | 13.968 | 14.282 | 14.717 | 15.058 | 15.366 | 16.901 | 18.448 |
| | Div-CLIP-L | 0.322 | 0.311 | 0.304 | 0.298 | 0.293 | 0.290 | 0.287 | 0.284 | 0.282 | 0.280 | 0.278 | 0.276 | 0.275 | 0.273 | 0.268 | 0.265 |
| | Div-Dinov2-L | 1.215 | 1.165 | 1.134 | 1.111 | 1.092 | 1.078 | 1.068 | 1.059 | 1.051 | 1.044 | 1.039 | 1.034 | 1.030 | 1.025 | 1.008 | 1.001 |
| | Div-CLIP-S | 0.326 | 0.314 | 0.307 | 0.301 | 0.296 | 0.293 | 0.290 | 0.287 | 0.285 | 0.283 | 0.282 | 0.279 | 0.278 | 0.277 | 0.272 | 0.269 |
| | Div-Dinov2-S | 1.266 | 1.217 | 1.186 | 1.163 | 1.145 | 1.132 | 1.120 | 1.110 | 1.104 | 1.097 | 1.093 | 1.088 | 1.081 | 1.078 | 1.063 | 1.054 |
| **gradient** | clip-score | 0.2546 | 0.2637 | 0.2682 | 0.2708 | 0.2726 | 0.2741 | 0.2751 | 0.2759 | 0.2769 | 0.2775 | 0.2782 | 0.2788 | 0.2793 | 0.2798 | 0.2812 | 0.2822 |
| | FID | 16.242 | 12.911 | 11.942 | 11.771 | 11.915 | 12.202 | 12.522 | 12.875 | 13.290 | 13.571 | 13.888 | 14.294 | 14.692 | 14.923 | 16.392 | 17.870 |
| | Div-CLIP-L | 0.323 | 0.311 | 0.304 | 0.298 | 0.294 | 0.290 | 0.287 | 0.285 | 0.283 | 0.280 | 0.278 | 0.277 | 0.275 | 0.274 | 0.268 | 0.265 |
| | Div-Dinov2-L | 1.218 | 1.168 | 1.137 | 1.112 | 1.095 | 1.081 | 1.070 | 1.062 | 1.053 | 1.045 | 1.039 | 1.034 | 1.029 | 1.026 | 1.009 | 1.000 |
| | Div-CLIP-S | 0.327 | 0.315 | 0.308 | 0.302 | 0.297 | 0.294 | 0.290 | 0.288 | 0.286 | 0.284 | 0.282 | 0.280 | 0.279 | 0.277 | 0.272 | 0.269 |
| | Div-Dinov2-S | 1.269 | 1.220 | 1.187 | 1.165 | 1.147 | 1.134 | 1.121 | 1.111 | 1.104 | 1.097 | 1.091 | 1.086 | 1.082 | 1.076 | 1.061 | 1.050 |

Table 6: **Experiment on Motion Latent Diffusion**, we present MMdist, FID and MultiModality, for (a) baseline, (b) heuristic linear, (c) cosine, and (d) gradient-based. We observe that the *gradient-based* method outperforms other in terms of FID-MMDist and diversity balance, with *cosine* reaching marginally worse FID vs. MMDist. and diversity performance. The **linear** method achieves a better trade-off than the baseline, however, worse than the *cosine* and *gradient-based*.

| | w | 1 | 2 | 4 | 6 | 8 | 9 | 14 | 19 |
|---|---|---|---|---|---|---|---|---|---|
| **baseline** | MMDist | 3.781 | 3.503 | 3.327 | 3.284 | 3.283 | 3.293 | 3.391 | 3.526 |
| | FID | 0.343 | 0.353 | 0.389 | 0.415 | 0.440 | 0.453 | 0.559 | 0.743 |
| | MultiModality | 3.731 | 3.208 | 2.789 | 2.616 | 2.544 | 2.537 | 2.592 | 2.704 |
| **linear** | MMDist | 4.350 | 3.743 | 3.502 | 3.411 | 3.380 | 3.375 | 3.392 | 3.284 |
| | FID | 0.406 | 0.299 | 0.306 | 0.336 | 0.382 | 0.408 | 0.358 | 0.412 |
| | MultiModality | 4.599 | 3.685 | 3.214 | 2.994 | 2.879 | 2.843 | 2.926 | 2.802 |
| **cosine** | MMDist | 4.443 | 4.181 | 3.874 | 3.708 | 3.610 | 3.575 | 3.479 | 3.450 |
| | FID | 0.424 | 0.355 | 0.295 | 0.280 | 0.282 | 0.287 | 0.326 | 0.392 |
| | MultiModality | 4.730 | 4.367 | 3.910 | 3.632 | 3.434 | 3.361 | 3.129 | 3.009 |
| **gradient** | MMDist | 4.309 | 4.008 | 3.706 | 3.563 | 3.484 | 3.456 | 3.397 | 3.393 |
| | FID | 0.392 | 0.325 | 0.288 | 0.289 | 0.299 | 0.307 | 0.372 | 0.459 |
| | MultiModality | 4.543 | 4.114 | 3.624 | 3.342 | 3.173 | 3.109 | 2.910 | 2.832 |

Subsequently, participants were prompted with three questions for each comparison:

1. Which set of images is more realistic or visually appealing?
2. Which set of images is more diverse?
3. Which set of images aligns better with the provided text description?

In total, each participant responded to 90 questions. We analyzed the results by examining responses to each question individually, summarizing the collective feedback.

The feedback from 52 participants (27 for Stable Diffusion and 25 for Wuerstchen) is summarized in Figure 9 and Figure 10. Evidently, users have a clear preference for our progressive weighting methods over the static weighting scheme across the dimensions of realism and diversity. For the text-alignment, users prefer progressive weightings text alignment capacity in stable diffusion than baseline, whereas preferred baseline for Wuerstchen.

More specifically, for the study on *Stable Diffusion*: regarding realism, both gradient-based (*gn*) and cosine-based methods garnered over 70% of the preference votes. When assessing text alignment, the gradient-based method stood out, receiving over 60% score, the highest among the three methods. As for diversity, the *cosine* method clearly led the test, capturing more than 90% of the participant's preference over the baseline method.

Regarding the *Wuerstchen* study: given the similarity in curve shape between the gradient-based and linear methods, user feedback for both was similar too. Approximately 60% of users favored the realism of images generated using our method over the baseline. In terms of diversity, all our methods were favored, recording a preference rate of 70%. Nonetheless, when evaluating text alignment, users perceived all our proposed methods as inferior to the baseline. We attribute this to over-diverse results that obfuscate the semantic significance inherited in the image.

## A.5 QUALITATIVE RESULTS

**Stable Diffusion v1.5.** In this section, we show some qualitative results: in Figure 13 and Figure 14 showed qualitative results of using our methods: *linear*, *cosine* and *gradient-based* progressive weighting compared against baseline. It sees clearly that the proposed method generates more diversity which the baseline suffers from collapsing problem, i.e., different sampling of same prompt seems only generate similar results. In some figures, e.g., Figure 13 with prompt: *"Two birds fighting high in the sky with bad weather."* we can see that the baseline ignores the rainy weather condition given by input and our methods can correctly retrieve this information and illustrate in the generated images. Another example can be found in the prompt: *"A flower sitting in a vase in front of a bottle."*, that only our methods are able to depict both the *vase* and the *bottle* with much richer

diversity and image composition. However some negative examples can also be found in Figure 14, in particular the facial area of the teddy bear and feet of horse in the prompt: person riding a horse while the sun sets. We posit the reason of these artefacts are due the overmuting of the initial stage and overshooting the final stage during the generation (more discussion in Section 6).

**Wuerstchen.** We further illustrate qualitative results using the Wuerstchen model (Pernias et al., 2023) in Figure 15 and Figure 16. It is evident that the proposed progressive weighting yields enhanced diversity and detail in the generated images. However, as highlighted in the limitations section 6, over-attenuating the initial stages might introduce structural anomalies. These are particularly visible in the results generated using the *cosine* method. This phenomenon can be attributed to the fact that the initial stages of the Wuerstchen model inherently encapsulate more condition-related information than the Stable Diffusion models (as evidenced by the $g(t)$ plot in Figure 4). This observation also provides insight into the superior performance of the *linear* heuristic over the *cosine* one, since the former closely resembles the profile of the probed $G(t)$.

**Artefacts and Limitations.**

### A.6 PSEUDO CODE TO COMPUTE GRADIENT-BASED PROGRESSIVE WEIGHTING

---
**Algorithm 1** Computing the Gradient-based Weighting Curve

---
**repeat**
    $t \leftarrow U(0, 1000)$                 ▷ uniform sampling $t$ for timestep
    $g(t, c)[s] \leftarrow \|\nabla_c \|\epsilon_\theta (\mathbf{x}_t, \mathbf{c}) - \boldsymbol{\epsilon}\|_1 \|_2$    ▷ computing L2 norm of gradient of condition to
sample on L1 loss

**until** K steps                         ▷ total steps of probing
$g(t, c) \leftarrow \sum_{s=0}^{S} g(t, c)[s]/S$       ▷ average over $S$ samples for smoothing reason
$G(t, c) \leftarrow \sum_{i=t}^{T} g(i, c)$                ▷ inverse cumulative
$G(t) \leftarrow \sum_{c=0}^{C} G(t, c)/C$         ▷ average over $C$ condition labels
$w(t) \leftarrow G(t)/\bar{G}(t)$            ▷ standardize by dividing the average

---

In this section, we present the pseudo-code detailing the computation of our gradient-based progressive weighting. Initially, we gather the L2 norm of the gradient with respect to the condition, which is derived from the L1 loss during the training. This step is executed over $K$ iterations for better alleviate the stochasticity and cover all timesteps. Subsequently, we calculate the expectation and employ an inverse cumulative distribution function based on the collected gradnorm data, before the final standardization process.

### A.7 COMPARISON WITH PARAMETERIZED POWER-COSINE-4

We test our results with those of Gao et al. (2023), which employs an empirically parameterized function based on power-cosine. While they optimized this function specifically for their dataset and method (CIN256 Deng et al. (2009)), adopting the parameter value of 4 as reported in their paper, Figure 17b (tagged pcs4) indicates that their dedicated optimization process yields superior results on their target dataset: CIN256. However, when this meticulously optimized curve is applied to Stable Diffusion, it significantly underperforms, as evident in Figure 17b, with both FID and CLIPScore seeing marked degradation. This phenomenon underlines our argument that the optimal curve can vary across datasets and methods. Our gradient-based approach offers valuable insights in this regard, pointing towards a more versatile solution, even if it is not the optimal.

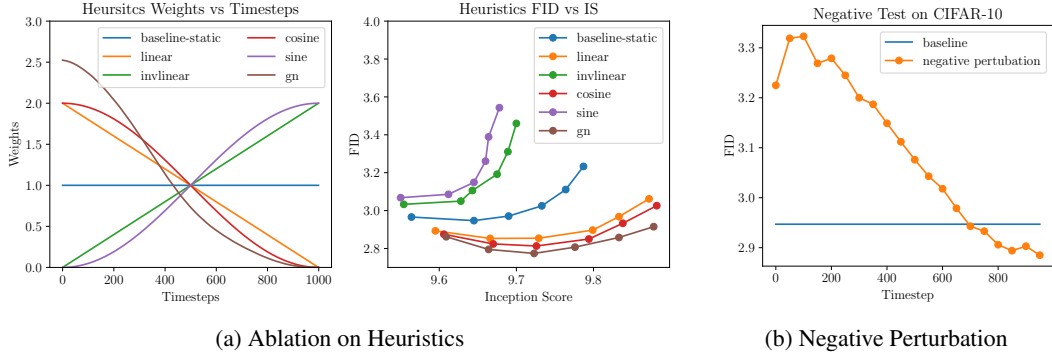

(a) Ablation on Heuristics         (b) Negative Perturbation

Figure 7: **Ablations Studies.** (a) Various heuristic (standardized) vs a gradient-based *gn* curves with their corresponding FID vs. IS performances. (b) Negative perturbation by setting the guidance scale to 0 across distinct intervals while preserving the default static guidance scale to the rest. A marked elevation in FID around the $t=100$ interval underscores its pivotal role in high image quality, as removing this interval leads to a worse FID compared to the baseline. By eliminating the weight at the initial stage, the FID experiences an enhancement.

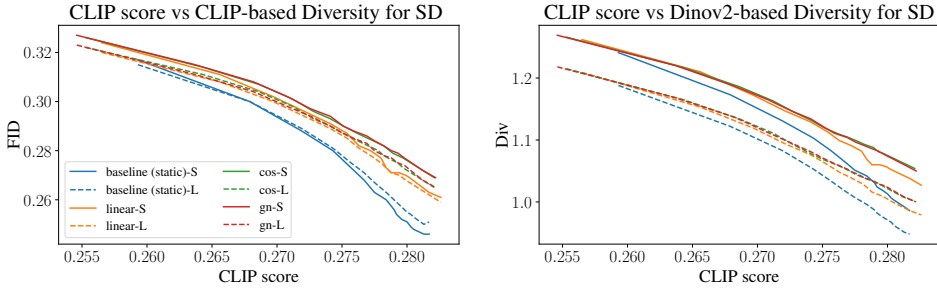

Figure 8: **Experiment on Stable Diffusion on two types of diversity.** Zero-shot COCO 10k CLIP-Score vs. Diversity computed by CLIP and Dino-v2 respectively.

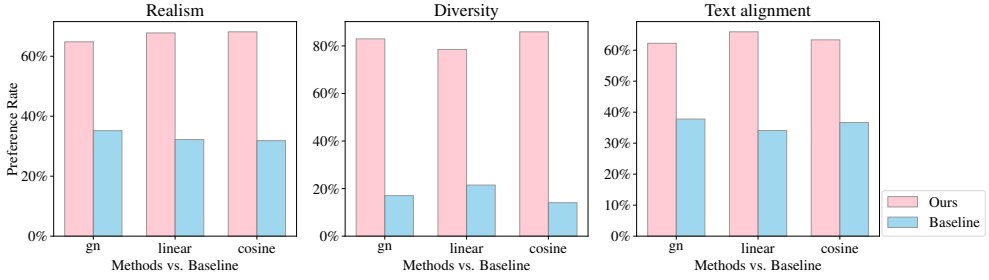

Figure 9: **User Study on Stable Diffusion.** The user study reveals that images generated with the proposed progressive weighting schemes are consistently preferred to ones from the baseline in terms of realism ($\sim 70\%$), text alignment ($\sim 60\%$), and especially for diversity (over $80\%$).

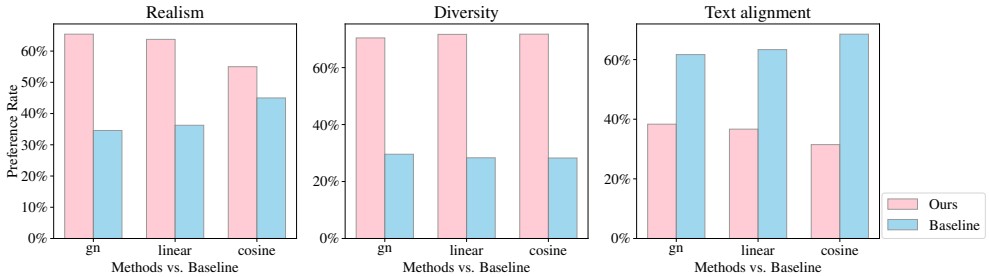

Figure 10: **User Study on Wuerstchen.** The user study reveals that images generated with the proposed progressive weighting schemes are consistently preferred to ones from the baseline in terms of realism ($\sim 60\%$), diversity (over $70\%$) but reported lower text-image alignment ($\sim 60\%$), especially for cosine heuristic.

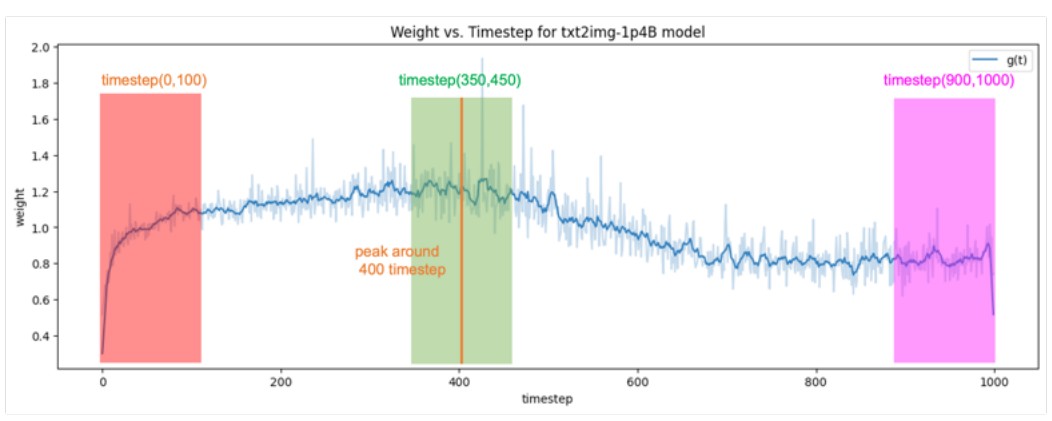

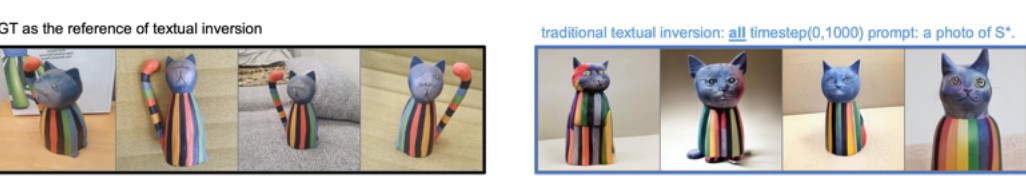

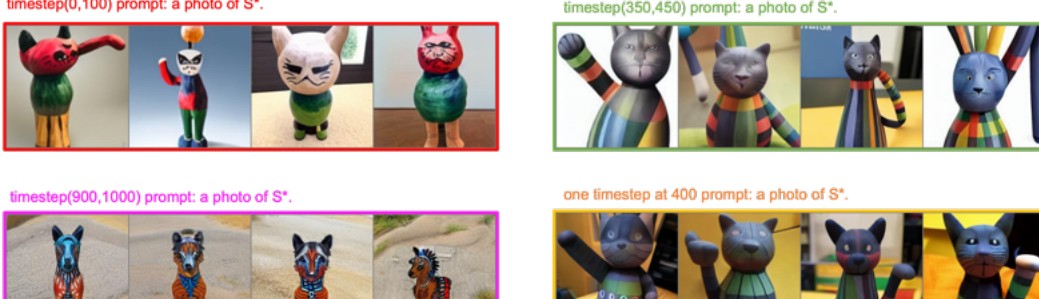

Figure 11: **Textual Inversion at Different Timesteps**: we show that by sampling on different timesteps ranges, the generated results are highly relevant to the gradient of condition response $g(t)$ showcased at the top of the figure. This supports the main argument that the condition information's influence is different across all generation process and the gradient of condition response $g(t)$ can be served as an effective measure to this information.

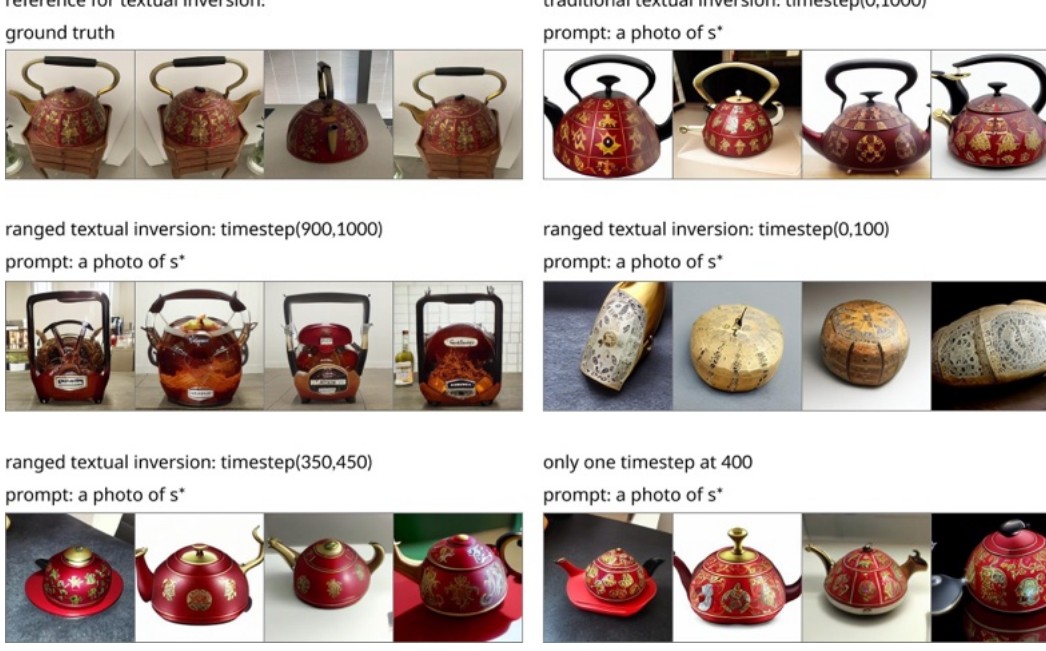

Figure 12: Another example of doing textual inversion by sampling *all* timesteps or ranged timesteps at different stage, it sees clear that by only using the timestep around the peak of the gradient information showed in Figure 12, the computed results shows more perceptual similarity to the ground truth (left-top panel) than using the timestep range with lower gradient norm area.

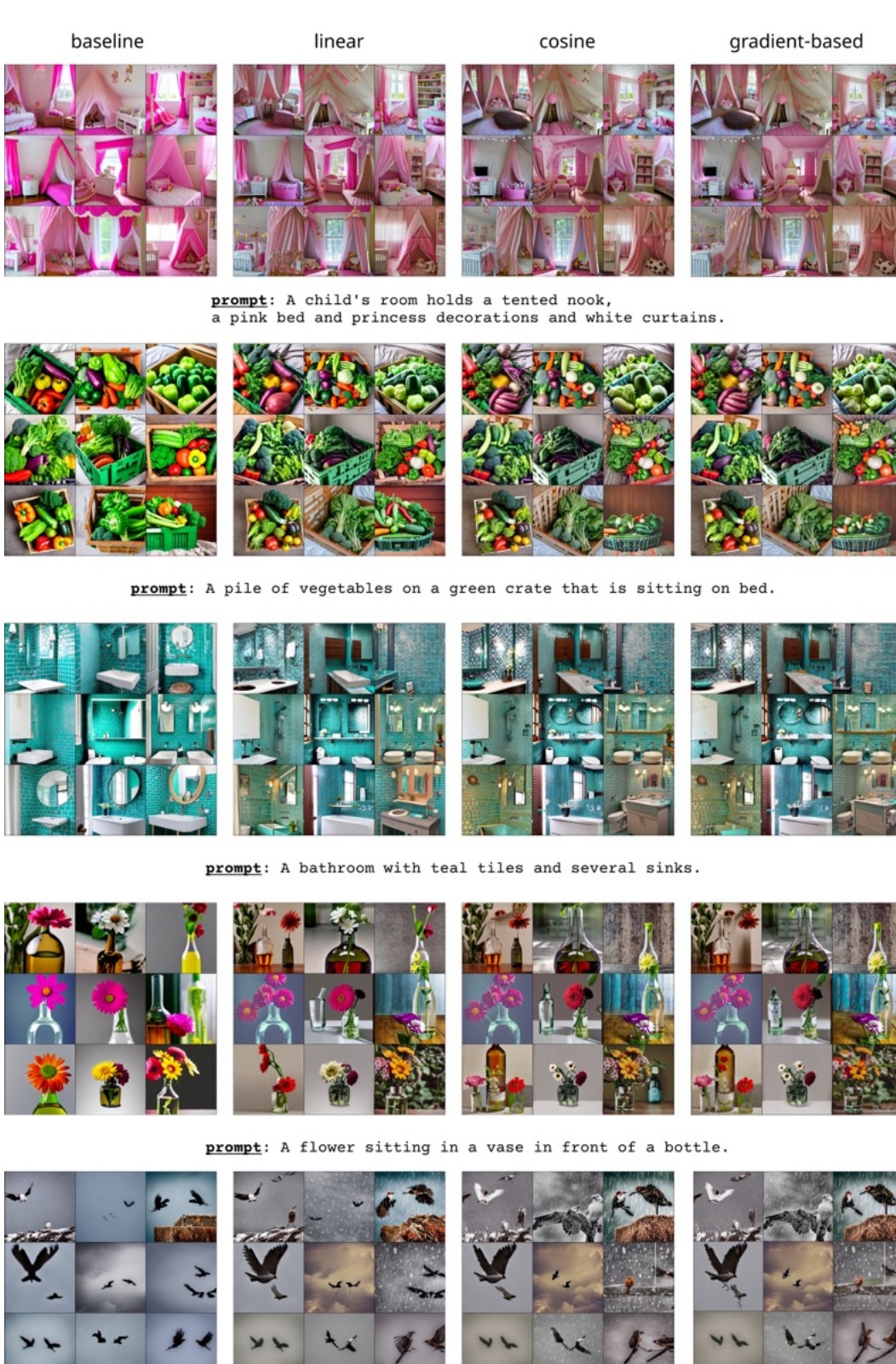

Figure 13: Qualitative evaluations of our proposed methods (linear, cosine, and gradient-based) against the baseline from Stable-Diffusion v1.5. For each textual prompt displayed beneath images, nine images are generated.

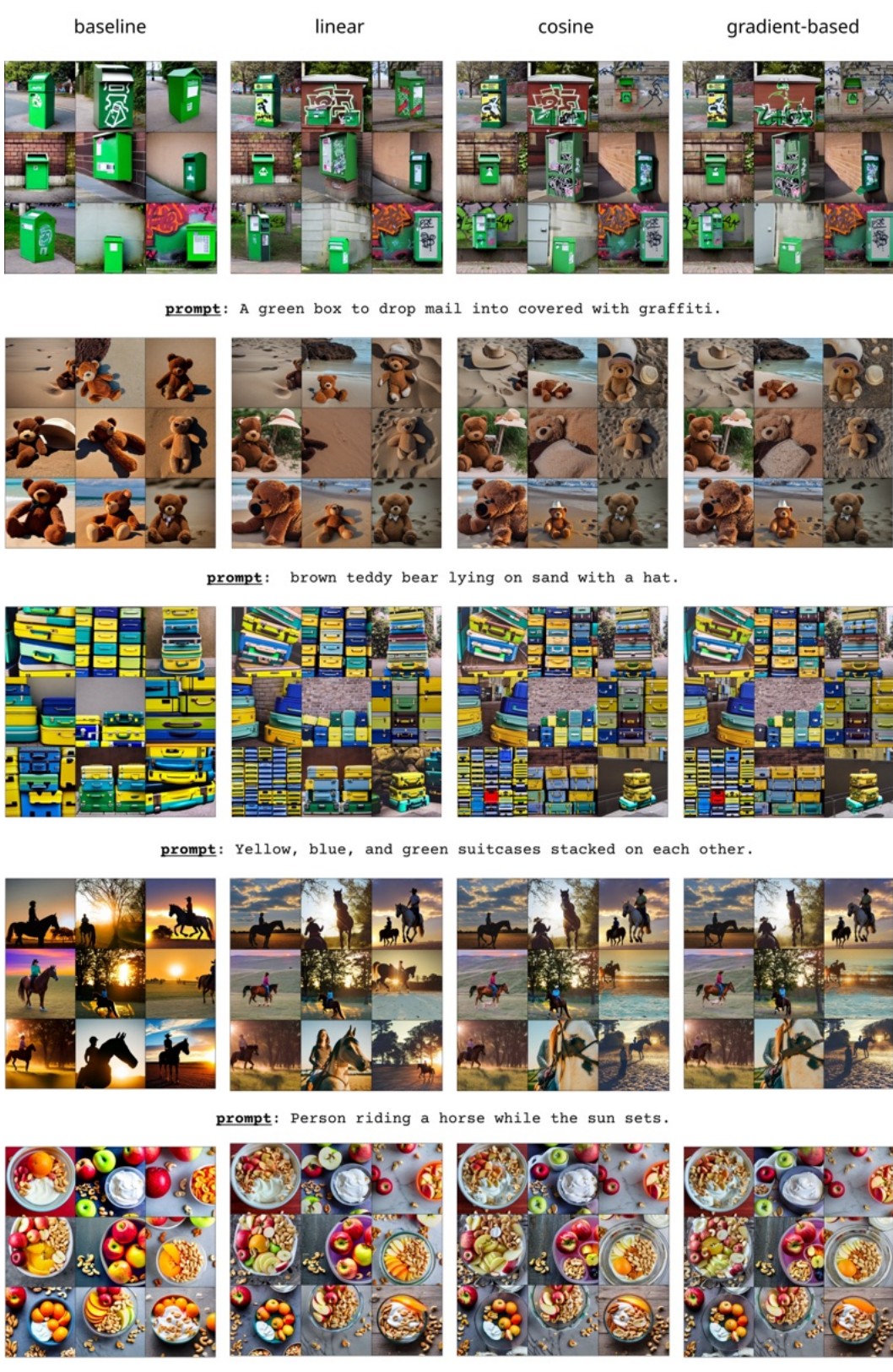

Figure 14: Qualitative evaluations of our proposed methods (linear, cosine, and gradient-based) against the baseline from Stable-Diffusion v1.5. For each textual prompt displayed beneath images, nine images are generated.

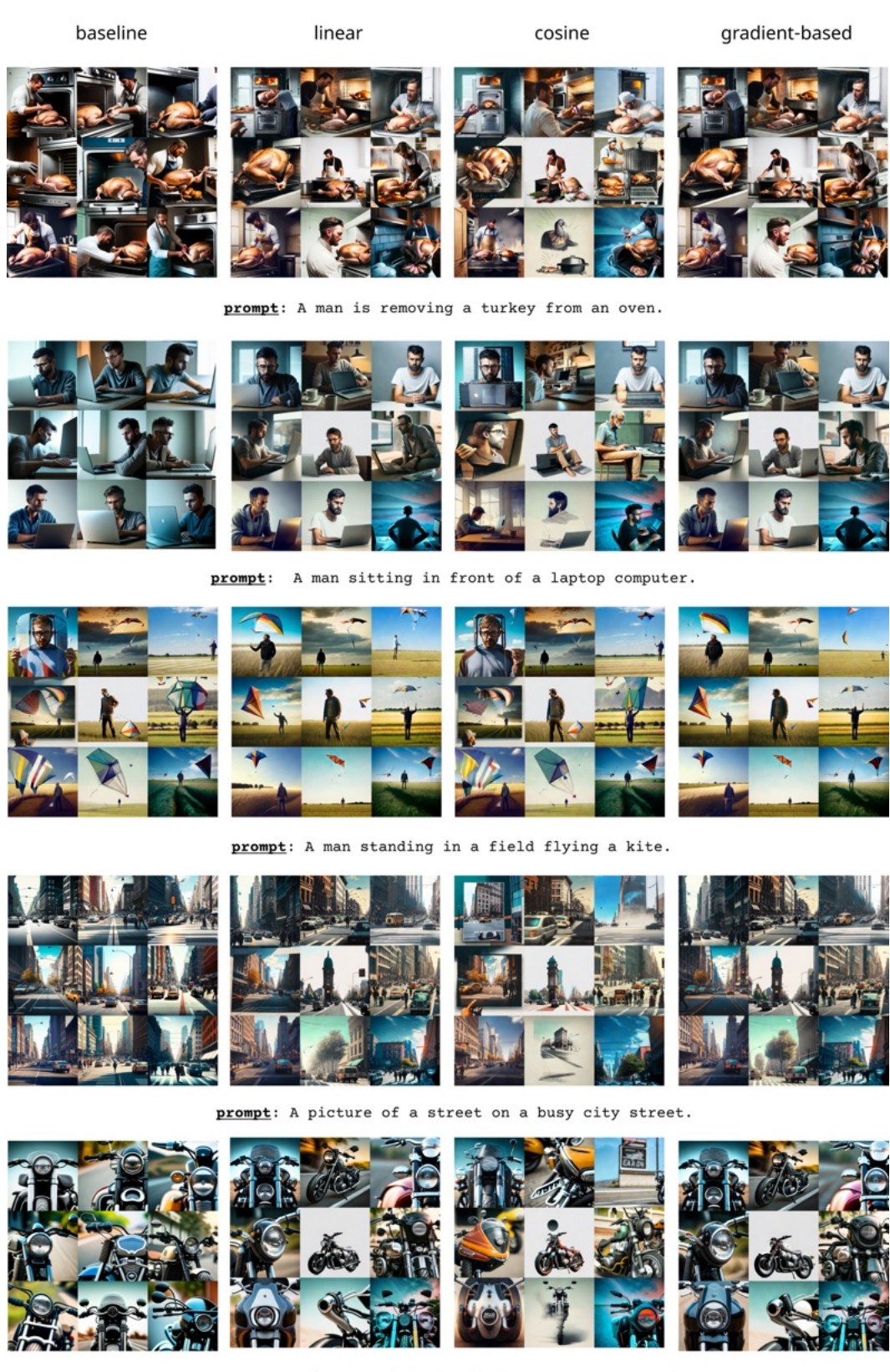

Figure 15: Qualitative evaluations of our proposed methods (linear, cosine, and gradient-based) against the baseline from Wusrtschen. For each textual prompt displayed beneath images, nine images are generated.

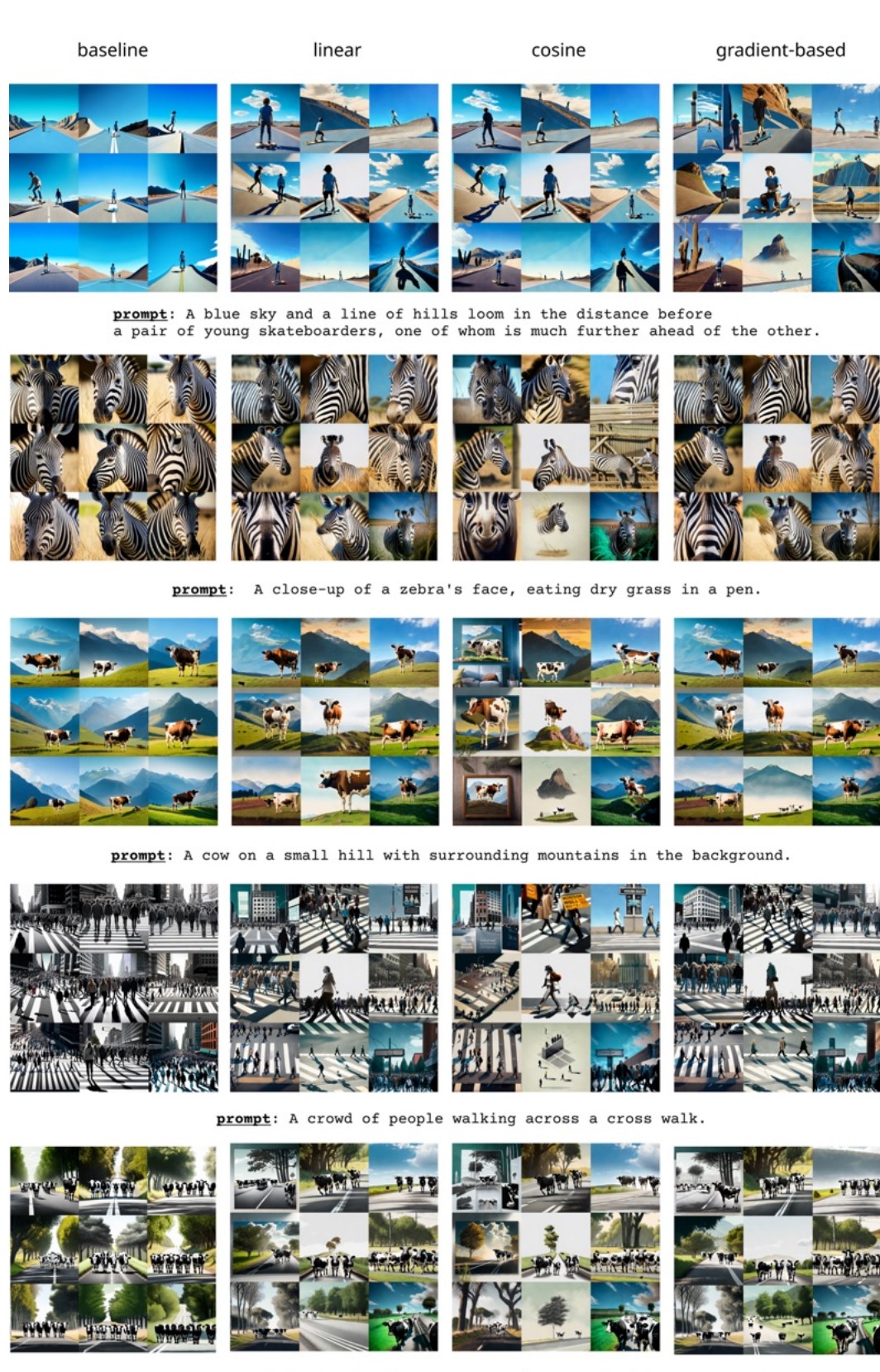

Figure 16: Qualitative evaluations of our proposed methods (linear, cosine, and gradient-based) against the baseline from Wusrtschen. For each textual prompt displayed beneath images, nine images are generated.

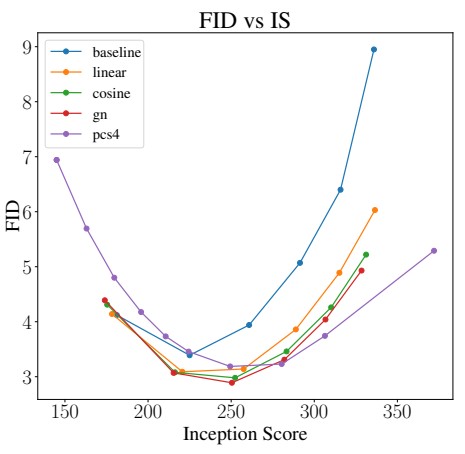

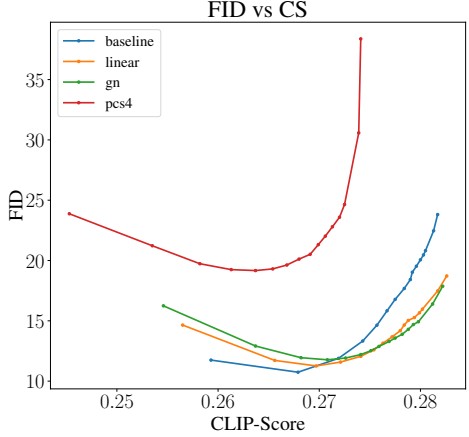

(a) **Comparison with Gao et al. (2023) in CIN256**

(b) **Comparison with Gao et al. (2023) in Stable Diffusion v1.5**

