# OpenReview forum: "To guide or not to guide: Improving diffusion sampling with progressive guidance"
_ICLR.cc/2024/Conference — ICLR 2024 Conference Withdrawn Submission_

### Official Review · Reviewer_y49U · 2023-10-28

**Soundness:** 2 fair
**Presentation:** 2 fair
**Contribution:** 2 fair
**Rating:** 3
**Confidence:** 4

**Summary:**

Diffusion guidance involves using conditional diffusion with specific constraints, like class labels or textual inputs. There are two main methodologies: Classifier Guidance, which uses a pre-trained classifier to guide generation, and Classifier-Free Guidance, which achieves implicit guidance without a classifier by incorporating the condition during training. In this paper, they propose a progressive weighting scheme called Progressive-Guidance, where the weight of the guidance term depends on the timestep. They demonstrate improved performance on benchmark metrics for three tasks: class-conditional image generation, text-to-image generation, and text-to-motion generation.

**Strengths:**

- The method is easy to implement and has been validated in three different application scenarios.
- The experiment covers a wide range of perspectives on diffusion models.

**Weaknesses:**

- There is no theoretical foundation for the statement, "gradient direction received from the guidance term that may conflict with the generation term, i.e., the force that brings the process towards the two Gaussian distributions." However, in Yang Song's SDE paper, it has been shown that epsilon prediction is related to the score prediction delta logp(x). The classifier-guidance is designed based on a simple Bayesian equation between log p(x) and log p(y|x). The strength of the guidance can be seen as a temperature factor for p(x)[p(y|x)]^w. The baseline is fully grounded in theory, while the author only provides empirical exploration in different tasks, which contradicts the mathematical theory behind it.
- The paper requires further polishing as the images appear blurry. Additionally, the conclusion and contribution sections lack clarity.
- Most importantly, the related works section is completely missing. There are numerous relevant works on the guidance of diffusion models, including discussions on noise schedule, guidance type, and guidance annotation dependency. The absence of these discussions makes it difficult to distinguish the proposed methods from others.

**Questions:**

as above

---

### Official Review · Reviewer_RdGV · 2023-10-29

**Soundness:** 3 good
**Presentation:** 2 fair
**Contribution:** 3 good
**Rating:** 6
**Confidence:** 4

**Summary:**

This paper proposes to make the guidance scale dependent on the time step for the guided diffusion sampling process. Two hand-crafted weighting schedules are proposed first. Then an adaptive weighting schedule is further proposed based on the gradient norm.

**Strengths:**

1. The proposed method is well-motivated based on empirical findings and previous studies on the limitations of a constant scale guidance.

2. The proposed method is simple and intuitive. The proposed gradient norm-based guidance is adaptive. The standardization removes the effect of hyper-parameters. The proposed method is elegant and potential to be easily applied.

3. The experiments include different forms of conditions, i.e., class-to-image, text-to-image, and text-to-motion. Experiments and abaltions studies are comprehensive in general.

4. The toy experiment is illustrative.

**Weaknesses:**

1. As for the guidance based on the gradient norm, the current solution only depends on the influence 'intensity' of the condition but not the 'correctness' of such influence. Above Eq.5, why does a lower influence mean a less reliable classifier? The intensity of the influence has no direct relation to the correctness of the influence.

2. Based on the above point, I think the proposed gradient-norm guidance might have a different interpretation. It provides more guidance for time steps that really need the guidance, but whether the guidance is really useful or not is not in the scope.

3. All the figures are in low resolution, especially the text in the figure.

4. For Figure 5, it is better to annotate the scale on the graph. Also, will the same trend still exist for a larger range of guidance scales?

5. Figure 6 shows little differences between the images of naive guidance and adaptive guidance.

**Questions:**

Please refer to Weakness 1 and 4.

---

### Official Review · Reviewer_4fCg · 2023-10-29

**Soundness:** 3 good
**Presentation:** 3 good
**Contribution:** 2 fair
**Rating:** 3
**Confidence:** 5

**Summary:**

In this paper, the authors propose a method to improve diffusion sampling with progressive guidance, which adjusts the weight of the conditional term according to the timestep. In particular, an interesting empirical analysis is conducted and attributes the potential conflicts between image quality and condition to the misclassification and conflicted gradients from the classifier in the initial stages of generation. To address this problem, the authors propose progressive guidance by using a time-step-dependent weighting scheme without retaining the models.

**Strengths:**

1.	The empirical analysis in section 3 is interesting and reasonable.
2.	The motivation of this paper is clear.

**Weaknesses:**

1.	The main idea of the proposed methods is to attenuate the influence of guidance in the initial stages. However, based on my experiences, with small guidance weight in the initial stages, the structural integrity and layout of the generated images by diffusion models would be disrupted, particularly for some complex structures. The authors should provide more evaluations of the samples with complex structures, such as instances with human bodies or longer prompts.  Actually, I believe that the trade-off between sample fidelity and conditional adherence still exists, even with the proposed methods.
2.	I have read the appendix carefully. It is weird to find that the proposed methods cannot outperform the baselines in terms of CLIP-Score with a small guidance weight w.  If I focus on the experimental results with w<10, which is actually used in practice, what are the corresponding trade-off curves in Figure 5?
3.	The paper does not conduct extensive experiments on different model architectures, and samplers to validate the robustness and generality of their method.
a)	Models: some more powerful diffusion models should be involved for comparisons, e.g., SD XL[1], and DeepFloyd IF[2], which can both be accessed publicly. Based on my understanding, those models can provide a more accurate estimation for classifications at the initial stage of the generation. As a result, the performance improvement of the proposed methods may decrease in those models.
b)	Samplers: DDIM sampler is the main selection for this paper to sample images during generation. More samplers, such as DPMSlover[3], should also be considered.
c)	Human-level metric: It is well-known that some quantitative metrics like FID, and IS, may be problematic in some cases. Human evaluation should be involved to provide more clear comparisons.
[1] https://huggingface.co/stabilityai/stable-diffusion-xl-base-1.0
[2] https://huggingface.co/DeepFloyd/IF-I-XL-v1.0
[3] Dpm-solver: A fast ode solver for diffusion probabilistic model sampling in around 10 steps

**Questions:**

1.	What are the corresponding trade-off curves in Figure 5 if we focus on the cases with guidance weight w<10.0?
2.	Provide more experimental analysis mentioned above to validate the proposed methods.

---

### Official Review · Reviewer_fKi6 · 2023-10-31

**Soundness:** 3 good
**Presentation:** 3 good
**Contribution:** 3 good
**Rating:** 6
**Confidence:** 5

**Summary:**

The manuscript provides a way to adapt the weight for guidance sampling in the diffusion model based on the two hypothesis misclassification and conflicted gradients. The adaptive scheme is conducted via heuristics such as linear, cosine, or gradient approaches. The results offer some improvements compared to the baseline.

**Strengths:**

The paper works on an important problem in diffusion sampling. The guidance scheme seems to boost the performance of the diffusion models, yet the current understanding of this method is still limited. The work shows that some flaws of guidance are due to the misclassification of the classifier as well as the conflicted gradients. While the literature has shown the conflicts between gradients, this work re-affirm the topic and further solves the problem via a progressive weight scheme.

**Weaknesses:**

Although the work is interesting, there are still some concerns about the work:

1. An adaptive weighting scheme has been investigated by Zheng (2022), which discusses a way to adapt weight guidance to avoid the gradient vanishing of the classifier. Interestingly, based on the heuristic approach of the submission, with w(t) = (1 - t/T), at the end of the sampling steps, the weight is larger. This results in a similar observation in Zheng (2022). The main concern is whether the hypothesis proposed in this paper is actually validated, or it just adapts to reduce the gradient vanishing.
2. Since the scheme for adapting weight during sampling is also investigated by Zheng(2022), a comparison is required.
3. The work also mentions the conflicted gradients which align with observations in Dinh et.al (2023). Is there a way to measure the conflicts after applying the proposed methods? Since the authors did not project gradients, there might be still some conflicts after the application of the progressive scheme.
4. The misclassification might not be well explained. When the image is totally noisy, there would be no information to classify. The main aim of the classification is to provide information for constructing the image. So what kind of information is misclassified? Do the authors mean that the information given by the classifier is wrong or the information given by the diffusion model is wrongly classified?
5. In Eq.(4) and (5), how is $\nabla _c \epsilon _{\theta}$ is calculated in the case of classifier-free guidance?
6. The meaning of Figure 3 is not so clear. Please revise the caption or Figure to clearly state the main purpose




Zheng, G. et al. (2022). Entropy-Driven Sampling and Training Scheme for Conditional Diffusion Generation. In: Avidan, S., Brostow, G., Cissé, M., Farinella, G.M., Hassner, T. (eds) Computer Vision – ECCV 2022. ECCV 2022. Lecture Notes in Computer Science, vol 13682. Springer, Cham. https://doi.org/10.1007/978-3-031-20047-2_43

**Questions:**

The main question is related to the writing and some motivations of the work. Please solve the concerns then I will raise the score.